# Deep-embedded clustering by relevant scales and genome-wide association study in autism

**Fumihiko Ueno**[1,2☯], **Ippei Takahashi**[2☯], **Hisashi Ohseto** [2*☯], **Tomomi Onuma**[1,2], **Akira Narita**[1], **Taku Obara**[1,2,3], **Mami Ishikuro**[1,2], **Keiko Murakami**[1,2], **Aoi Noda**[1,2,3], **Fumiko Matsuzaki**[1,2], **Hirohito Metoki** [1,2,4], **Gen Tamiya**[1,2,5], **Shigeo Kure**[1,2,6], **Shinichi Kuriyama** [1,2,7*]

**1** Tohoku Medical Megabank Organization, Tohoku University, Sendai, Japan, **2** Graduate School of Medicine, Tohoku University, Sendai, Japan, **3** Tohoku University Hospital, Tohoku University, Sendai, Japan, **4** Faculty of Medicine, Tohoku Medical and Pharmaceutical University, Sendai, Japan, **5** RIKEN Center for Advanced Intelligence Project, Tokyo, Japan, **6** Miyagi Children's Hospital, Sendai, Japan, **7** International Research Institute of Disaster Science, Tohoku University, Sendai, Japan

☯ These authors contributed equally to this work.
* hisashi.ohseto.b3@tohoku.ac.jp (HO); shinichi.kuriyama.e6@tohoku.ac.jp (SK)

## Abstract

Autism spectrum disorder (ASD) presents with heterogeneous phenotypic and genetic characteristics. Despite investigation into the molecular mechanisms underlying ASD, its etiology remains elusive. In our previous investigation within the Simons Simplex Collection (SSC), we noted increased signals through a genome-wide association study (GWAS) by clustering patients with ASD and reducing the sample size. This study seeks to validate our previous study in a different population, the Simons Foundation Powering Autism Research for Knowledge (SPARK) population, while probing further into the genetic architecture of ASD. We examined data from 2,079 white male subjects and 875 unaffected SPARK siblings. Our methodology encompassed cluster analyses, followed by traditional GWAS and cluster-based GWAS (cGWAS). No significant associations were observed in the conventional GWAS when comparing all patients with all controls. However, in the cGWAS, by comparing patients clustered by phenotypes with controls, we identified 27 chromosomal loci meeting the criteria of $p < 5.0 \times 10^{-8}$. Remarkably, several of these loci were situated within or in proximity to genes previously implicated as candidates for ASD. Nonetheless, our previous study of the SSC population did not fully replicate the SPARK population. The absence of reproducibility suggests the possibility of false positives within the cGWAS results due to potential technical factors. However, the emergence of multiple signals post-clustering and the association of numerous identified gene regions with ASD and related disorders provide supporting evidence for the validity of cGWAS outcomes.

**Data availability statement:** The data used in this study are available only to those who are granted access by the Simons Foundation due to restrictions related to participant privacy and ethical considerations. The Simons Foundation's data access policy ensures that sensitive genetic and phenotypic data of participants are protected in accordance with ethical guidelines and data protection regulations. Researchers who wish to access the data must submit a request to the SFARI Data and Biospecimen Repository (SDBR). Data access requests can be directed to the SDBR at sdbr@simonsfoundation.org.

**Funding:** The present study was supported by the Ministry of Education, Culture, Sports, Science and Technology (MEXT) KAKENHI Grant Numbers 19H03894, 21K17304 and 22H03346. The funders had no role in study design, data collection and analysis, decision to publish, or preparation of the manuscript.

**Competing interests:** The authors have declared that no competing interests exist.

## Introduction

Autism spectrum disorder (ASD) is a neurodevelopmental disorder primarily characterized by communication difficulties and repetitive behaviors [1]. Despite efforts to understand the molecular mechanisms underlying ASD, its etiology remains unclear [2]. Evidence suggests that genetic factors strongly contribute to the risk of ASD development [3]. For instance, identical twins exhibit a considerably higher ASD concordance rate of 92% compared with 10% in dizygotic twins [4]. Additionally, the risk ratio for ASD recurrence between siblings is reported to be 22 [5].

Previous genome-wide association studies (GWASs) have identified numerous genetic variants associated with ASD [6,7]. The link between these genetic variants and a single disease can be understood through a polygenic model, wherein the effect of each variant is small but collectively contributes to disease development [8,9]. In the GWAS of a disease, a larger sample size generally aids in identifying more signals, whereas reducing the sample size decreases the number of identified signals. If a GWAS is conducted with a specific sample size and no significant signals are found, it becomes increasingly challenging to identify any signals if the sample size is further reduced by dividing the patients. However, according to a simulation study, the power of a GWAS can be enhanced by dividing patients into more homogeneous populations, regardless of the sample size [10]. In such cases, it may be possible to identify certain signals by clustering patients with similar phenotypes and investigating genetic factors.

In our previous study, we detected more signals by clustering patients with ASD and decreasing the sample size [11]. This finding sharply contrasts the findings of several GWASs and requires careful interpretation and considerable follow-up examination. Thus, we sought to validate our findings using a different dataset in the present study. We aimed to explore the genetic structure of ASD by categorizing patients into clusters based on phenotypic variables and conducting a GWAS (specifically, cluster-based GWAS [cGWAS]), as conducted in our previous study. Moreover, we increased the sample size [12] and introduced a deep-embedded clustering (DEC) algorithm [13,14].

## Materials and methods

### Participants

This study adhered to the guidelines of the Declaration of Helsinki [15] and all other relevant guidelines. The Institutional Review Board of Tohoku University Graduate School of Medicine (2020-1-826) approved our protocol, and written informed consent was obtained from participants using the Simons Foundation Autism Research Initiative (SFARI) for the SPARK study, which began recruitment on 21/04/2016, and is ongoing [12]. The data were initially obtained on 03/12/2019, for a different study and were accessed for research purposes at 01/10/2021. In SPARK, phenotypic data and biospecimens were collected remotely, enabling participants to fulfill the study requirements online at a convenient time. Individuals in the United States with a professional ASD diagnosis, their parents, and unaffected siblings were eligible to

participate in SPARK. Phenotype information and ASD diagnoses in SPARK are self- or parent-reported, and the Interactive Autism Network suggests that parent-reported diagnoses of ASD are highly valid [16].

## Datasets

We used phenotypic variables, background history, and genotypic data from the SPARK database, which was publicly released in October 2017 and is directly available from SFARI [12]. From the SPARK WES1 (27 K) dataset, we used data from 2,685 affected white male probands for whom data from all three tests, i.e., Developmental Coordination Disorder Questionnaire (DCDQ) [17], RBS-R [18], and Social Communication Questionnaire (SCQ) [19], were available, and 891 unaffected male proband siblings for whom genotype data were generated by the Illumina Infinium Global Screening Array (GSA) v1.0. ASD is consistently more prevalent in males [2]. Additionally, sex-linked etiology and susceptibility have been reported in autism [20]. Therefore, we focused only on males to exclude sex-related heterogeneity. Of the 2,685 affected probands, we excluded 606 who were biologically related to their unaffected male siblings to eliminate bias due to familial relatedness. Consequently, 2,079 probands and 891 unaffected male siblings were eligible for further analysis. To exclude participants whose ancestries significantly varied, principal component analyses were performed on genotype data using EIGENSOFT version 7.2.1 [21]. Based on these analyses, we excluded 33 individuals whose data points exceeded six standard deviations for principal components 1 or 2. Therefore, 2,062 probands and 875 unaffected male siblings were eligible for clustering and GWAS.

## Clustering

We conducted cluster analyses using the phenotypic variables of DCDQ (15 items), SCQ (40 items), and RBS-R (43 items) scoring; age at initial registration in months; self-reported ethnicity; dominant hand; and history of medication, biomedical intervention (e.g., diet, alternative medicine, and supplements), and intensive behavioral intervention (e.g., Applied Behavior Analysis, Verbal Behavior, Pivotal Response Treatment). Missing data were imputed using the mean value of each variable, and all categorical data were transformed into dummy variables.

We applied DEC [13,14], which uses a deep-learning algorithm to conduct cluster analysis using phenotypic variables. The DEC algorithm requires the specification of several parameters, including the number of clusters (k), iterations, epochs, and network dimensions. For this study, we predetermined k to be 40, assuming that ASD consists of hundreds of subgroups [6] and considering the statistical power derived from sample-size calculations [22]. Other hyperparameters for DEC included a batch size of 256,300 pre-training epochs, 400 maximum iterations, 30 update intervals, and 0.001 tolerance for the stopping criterion. These analyses were performed using the scikit-learn toolkit in Python 2.7.

Clustering serves as a technique for exploratory data analysis where the validity of clustering outcomes can be assessed using external knowledge, such as the purpose of segmentation [23]. Several methods have been proposed to predefine the number of clusters (k), including visual examination, likelihood, and error-based approaches. However, it is noteworthy that these methods may not always yield mutually consistent results [24]. Although metrics exist for assessing the quality of clusters [25], the number of clusters should align with the research purposes. Therefore, the inflation factor ($\lambda$) of quantile-quantile plots using the logarithm of the $p$-value to base 10 ($-\log_{10}p$) for each cluster was calculated and used as validity indicators.

## Genotype data and quality control

We used the SPARK WES1 (27 K) genotypic dataset, in which the probands and unaffected male siblings were previously genotyped (https://gpf.sfari.org/hg38/datasets/SFARI_SPARK_WES_1_CONSORTIUM/dataset-description). We used the dataset genotyped by Illumina GSA v1.0 containing 642,824 probes. We excluded SNPs with a minor allele frequency < 0.01, call rate < 0.95, and Hardy–Weinberg equilibrium test $p < 0.000001$.

We independently imputed the SPARK WES1 (27 K) genotypic data from the phenotype data using the Michigan imputation server. The human genome reference build of the genotypic data was converted from hg38 to hg19 using LiftOver, a tool provided by the UCSC Genome Browser (http://genome.ucsc.edu/cgi-bin/hgLiftOver). On the Michigan imputation server, we selected "HRC r1.1 2016 (GRCh37/hg19)" for Reference Panel, "0.3" for Rsq Filter, "Eagle v2.4" for Phasing, and "Other/Mixed" for population options, and performed quality control and imputation.

After genotype imputation, the SPARK WES1 (27 K) genotypic dataset contained 33,717,335 SNPs on the autosomes.

## Statistical analysis

As a preliminary study, we conducted a conventional GWAS comparing all patients with all controls in the entire SPARK WES1 (27 K) genotypic dataset, with 2,062 male probands and 875 unaffected male siblings. The control group did not include the male siblings of the affected participants. Unaffected male siblings were selected from other proband families, including siblings of probands who did not respond to all survey forms and siblings whose female siblings were probands. In the second step, we conducted a cGWAS in each subgroup of cases, which were divided using the DEC algorithm [13,14] and controls. A logistic regression model was used to calculate the additive allele dosage effect.

The GWAS was performed using the PLINK software package [26]. The reported SNPs were annotated using ANNOVAR [27]. Manhattan plots were generated using R software (version 4.1.0; R Foundation for Statistical Computing, Vienna, Austria) [28]. For GWAS, no covariates were adjusted for in this study. Owing to the focus of the present study on males, the sex of children was not adjusted for, whereas age was not included as a covariate in the GWAS because it was used as a variable for clustering. Additionally, variables related to genetic architecture were not corrected for in the GWAS because populations with considerable genetic heterogeneity were excluded based on the SNP array data analysis via principal component analysis. The GWAS results obtained were clumped using linkage disequilibrium, and the locus with the lowest p-value was selected.

## Code availability

The computer code used to generate the results is available from the authors upon request. All computer code access inquiries should be sent to Shinichi Kuriyama (shinichi.kuriyama.e6@tohoku.ac.jp).

## Results

### Cluster-based GWAS

In a preliminary study, we conducted a conventional GWAS comparing all patients and controls using the SPARK WES1 (27 K) genotypic dataset and found no significant associations.

The DEC algorithm requires researchers to specify k clusters (k). The average inflation factor λ for a cGWAS with k = 40 was 0.9. Empirically, a threshold for λ considered safe to minimize the risk of false positives is a value less than 1.05 [29]. Therefore, we considered λ < 1.05 as an indicator of successful clustering. We considered cGWAS using cluster analysis with k = 40 as the most appropriate approach for the present dataset. Fixing the hyperparameter k of DEC to 40 eventually led to dividing the dataset into 39 clusters.

The characteristics of each cluster are presented as a heatmap in Table 1. For example, Cluster 5 had relatively high DCDQ [17], Repetitive Behaviors Scale-Revised (RBS-R) [18], average SCQ scores [19], no underlying disease, and a high rate of interventional treatment.

### Gene interpretation

We observed 27 chromosomal loci that satisfied the threshold of $p < 5.0 \times 10^{-8}$. The results of each GWAS analysis with a sample size of more than nine cases following clustering are shown in Table 2. Several loci were identified either within or

near the genes linked to the Human Gene module of the SFARI Gene scoring system [6], including RFX3 (score 1, Rare Single Gene Mutation, Syndromic) in Cluster 15; HCN1 (score S, Rare Single Gene Mutation, Genetic Association) in Cluster 18; CSMD1 (score 3, Rare Single Gene Mutation, Genetic Association) in Cluster 23; HIVEP3 (score 2, Rare Single Gene Mutation, Genetic Association) in Cluster 24; and CNTNAP2 (score 2S, Rare Single Gene Mutation, Syndromic, Genetic Association) in Cluster 30. The SFARI Gene scoring system ranges from "Category 1," which indicates "High Confidence," to "Category 3," which denotes "Suggestive Evidence." Genes associated with ASD-related syndromic disorders are classified under a distinct category, labeled "#S" (e.g., 2S and 3S). Meanwhile, rare single gene variants, disruptions/mutations, and submicroscopic deletions/duplications associated with ASD fall under the category of "Rare Single Gene Mutation."

Alongside genes from the Human Gene module of the SFARI Gene, our findings also included several other crucial genes previously reported to be associated with ASD and related disorders, listed as follows (Table 3): PLA2G4A in Cluster 28, STMN4 in Cluster 19, PMP22 in Cluster 24, and ADAM12 in Cluster 21, previously associated with ASD [37,47,55–57,61,67,69,76,84]; NCOR2 in Cluster 9, RFX3 in Cluster16, and CEP112 in Cluster 39, previously associated with attention deficit hyperactivity disorder [37,89,90]; UMAD1 in Cluster 17, HCN1 in Cluster 19, PACERR, and KCNAB1in Cluster 28, previously associated with epilepsy [45,46,75,80,81]; KCNAB1 in Cluster 28, previously associated with mental retardation [80]; ADAM12 in Cluster 21, previously associated with Down's syndrome [57]; PLA2G4A in Cluster 28, CSMD1 in Cluster 24, and ADAM12 in Cluster 21, previously associated with schizophrenia [57,63,64,77]; NCOR2 in Cluster 9 and TAFA5 in Cluster 23, previously associated with depressive disorder [39,59]; HCN1 and MRPS30 in Cluster 19, SMAD3 in Cluster 23, CSMD1 in Cluster 24, and HIVEP3 and NUBPL in Cluster 25, previously associated with Parkinson's disease [49,51,60,65,70,71,74]; GMPS in Cluster 28 and THBS2 and WDR27 in Cluster 8, CSMD1 in Cluster 24, ADAM12 in Cluster 21, SCARB1 in Cluster 9, AATF in Cluster 31, TAFA5 in Cluster 23, and HCN1 in Cluster 19, previously associated with Alzheimer's disease [33,35,40,49,57,62,79,83]; RPA3 in Cluster 17, previously associated with Machado–Joseph disease [42]; NUFIP2 in Cluster 39, previously associated with microcephaly [88]; NCOR2 in Cluster 9, previously associated with spinal muscular atrophy [38]; and PMP22 in Cluster 24 previously associated with neuropathy [68].

In addition, our findings incorporated some significant genes linked to ASD symptoms identified in previous studies (Table 3), including WDR27 in Cluster 8, previously associated with sleep disturbance [34], and HCN1 in Cluster 19, previously associated with post-traumatic stress disorder [48]. Furthermore, we observed signals in important genes associated with ASD pathways (Table 3), such as RGS3 in Cluster 24, encoding a regulator of G protein signaling [66]; NUBPL in Cluster 25, previously associated with mitochondrial disease [73]; and OR13F1 in Cluster 8, encoding an olfactory receptor [36].

We further observed signals in various genes known to be mutated in cancer (Table 3), including ETV1 in Cluster 7 and HIVEP3 in Cluster 25, previously associated with prostate cancer [31,72]; RFX3-AS1 in Cluster 16, MRPS30-DT and MRPS30 in Cluster 19, and MIR3201 in Cluster 23, previously associated with breast cancer [41,50,52,58]; GMPS in Cluster 28 and ZNF93 in Cluster 37, previously associated with ovarian cancer [78,86]; MIR548H4 in Cluster 19, AATF in Cluster 35, and APCDD1L-DT in Cluster 35, previously associated with lung cancer [53,82,85]; ARL4A in Cluster 7 and RPA3 in Cluster 17, previously associated with glioma [30,43]; ETV1 in Cluster 7, previously related to cranial germinomas [32]; MIR548H4 in Cluster 19, previously associated with head and neck squamous cell carcinoma [54]; RPA3 in Cluster 17, previously associated with gastric cancer [44]; and ZNF682 in Cluster 37, previously associated with Barrett's esophagus [87].

## Replication study

We previously conducted a cGWAS using a dataset from the SSC [11]. We considered the agreement between the SSC and SPARK results to be evidence of successful replication. After SSC data analysis, we found statistically significant single-nucleotide polymorphisms (SNPs) in the cGWAS for the following genes: CDH5, CNTN5, CNTNAP5, DNAH17,

**Table 1. Characteristics of the clusters.**

| Cluster no. | 1 | 2 | 3 | 4 | 5 | 6 | 7 | 8 | 9 | 10 | 11 | 12 | 13 | 14 | 15 | 16 | 17 | 18 |
|---|---|---|---|---|---|---|---|---|---|---|---|---|---|---|---|---|---|---|
| n | 195 | 6 | 16 | 2 | 80 | 262 | 21 | 38 | 11 | 6 | 195 | 10 | 168 | 32 | 193 | 52 | 63 | 11 |
| **DCDQ, mean** | | | | | | | | | | | | | | | | | | |
| Control during movement subscale score | 16.3 | 17.7 | 15.8 | 16.5 | 19.2 | 24.1 | 14.5 | 15.6 | 17.8 | 20.5 | 12.6 | 12.7 | 18.3 | 18.1 | 10.8 | 13.2 | 13.3 | 20.6 |
| Fine motor/handwriting subscale score | 8.9 | 10.3 | 8.6 | 5 | 11.7 | 15.1 | 10.1 | 6.7 | 8.5 | 9 | 5.5 | 8.6 | 11.8 | 11.4 | 5.8 | 7.6 | 7.5 | 11.5 |
| General Coordination subscale score | 11.2 | 11 | 11.1 | 12 | 11.4 | 16.8 | 9.1 | 8.4 | 10.6 | 11.2 | 9.7 | 8.2 | 12.2 | 13.4 | 7.8 | 8 | 8.2 | 10.1 |
| Total score | 36.5 | 39 | 35.5 | 33.5 | 42.5 | 56 | 33.5 | 30.6 | 36.9 | 40.8 | 27.8 | 29.5 | 42.4 | 42.9 | 24.3 | 28.8 | 28.7 | 42.3 |
| **RBS-R, mean** | | | | | | | | | | | | | | | | | | |
| Stereotyped behavior subscale score | 10.1 | 5 | 6.6 | 3.5 | 7.7 | 3 | 4.9 | 6.5 | 6.3 | 4.7 | 6.1 | 5.5 | 2.9 | 4 | 8.8 | 2.3 | 2.9 | 5.9 |
| Self-injurious behavior subscale score | 8.6 | 6.2 | 0.6 | 0 | 6.4 | 1.3 | 1.7 | 6.7 | 0.5 | 1.7 | 2.7 | 5.1 | 1.5 | 1 | 5.1 | 2.3 | 1.5 | 4.5 |
| Compulsive behavior subscale score | 11.3 | 4.2 | 4.6 | 2 | 7.6 | 2.4 | 4.9 | 6.1 | 8.3 | 4.2 | 3.4 | 2.8 | 2.3 | 5.4 | 9.2 | 2 | 4.1 | 4.4 |
| Ritualistic behavior subscale score | 10.8 | 6 | 6.9 | 1.5 | 9 | 2.9 | 5 | 8.8 | 7.5 | 8 | 2.6 | 6.1 | 3 | 6.3 | 9.5 | 3.3 | 6.3 | 6.5 |
| Sameness behavior subscale score | 18.2 | 7.7 | 7.5 | 3.5 | 15.8 | 4 | 7.2 | 14.2 | 10.2 | 10.2 | 4.7 | 7.8 | 4.1 | 9.1 | 16.7 | 5.8 | 9.3 | 10.2 |
| Restricted behavior subscale score | 7.5 | 3.8 | 3.9 | 2.5 | 6.2 | 2 | 3.7 | 6.7 | 6.3 | 3.2 | 3.8 | 3.8 | 2 | 4.3 | 6.7 | 2.8 | 3.1 | 4.7 |
| Total score | 66.6 | 32.8 | 30.1 | 13 | 52.7 | 15.7 | 27.5 | 49 | 39 | 31.8 | 23.4 | 31.1 | 15.7 | 30.2 | 56 | 18.4 | 27.2 | 36.1 |
| **SCQ, mean** | | | | | | | | | | | | | | | | | | |
| Total score | 27 | 22.8 | 20 | 24 | 23.9 | 17.8 | 22.5 | 27.6 | 24.4 | 27.7 | 23.6 | 27 | 14.1 | 18.6 | 29.6 | 21.7 | 19.7 | 22.5 |
| **Basic screening, %** | | | | | | | | | | | | | | | | | | |
| Premature birth | 12.8 | 33.3 | 0 | 0 | 10 | 8 | 4.8 | 13.5 | 0 | 0 | 9.7 | 50 | 6.5 | 25 | 18.2 | 3.9 | 17.7 | 0 |
| Macrocephaly | 6.7 | 0 | 0 | 0 | 0 | 2.3 | 0 | 5.4 | 9.1 | 0 | 7.2 | 0 | 1.2 | 0 | 7.3 | 5.9 | 8.1 | 0 |
| Seizure disorder or epilepsy | 6.2 | 16.7 | 0 | 0 | 1.2 | 3.4 | 0 | 10.8 | 0 | 0 | 7.2 | 20 | 2.4 | 0 | 12 | 2 | 3.2 | 9.1 |
| **Dominant hand, %** | | | | | | | | | | | | | | | | | | |
| Either or both | 24.7 | 16.7 | 31.2 | 0 | 12.5 | 9.2 | 4.8 | 28.9 | 18.2 | 0 | 28.1 | 10 | 13.1 | 9.4 | 21.8 | 3.8 | 4.8 | 9.1 |
| Left | 11.3 | 0 | 18.8 | 0 | 5 | 9.2 | 0 | 15.8 | 18.2 | 16.7 | 18.2 | 30 | 14.3 | 28.1 | 18.7 | 9.6 | 9.5 | 18.2 |
| Right | 63.9 | 83.3 | 50 | 100 | 82.5 | 81.7 | 95.2 | 55.3 | 63.6 | 83.3 | 53.6 | 60 | 72.6 | 62.5 | 59.6 | 86.5 | 85.7 | 72.7 |
| **Intervention, %** | | | | | | | | | | | | | | | | | | |
| Medication | 49.7 | 50 | 6.2 | 100 | 63.7 | 45.4 | 33.3 | 78.9 | 18.2 | 100 | 32.8 | 100 | 44 | 56.2 | 65.8 | 78.8 | 74.6 | 72.7 |
| Biomedical | 26.7 | 33.3 | 12.5 | 0 | 32.5 | 20.6 | 4.8 | 26.3 | 0 | 83.3 | 27.2 | 0 | 13.1 | 31.2 | 31.6 | 11.5 | 27 | 36.4 |
| Intensive behavioral | 52.8 | 33.3 | 37.5 | 50 | 41.2 | 39.7 | 14.3 | 44.7 | 27.3 | 50 | 51.3 | 0 | 34.5 | 21.9 | 52.8 | 30.8 | 25.4 | 81.8 |

DCDQ, Developmental Coordination Disorder Questionnaire; RBS-R, Repetitive Behavior Scale-Revised; SCQ, Social Communication Questionnaire; SD, standard deviation; NA, not applicable. High values among clusters are shown in red, and low values in blue.

| 19 | 20 | 21 | 22 | 23 | 24 | 25 | 26 | 27 | 28 | 29 | 30 | 31 | 32 | 33 | 34 | 35 | 36 | 37 | 38 | 39 |
|---|---|---|---|---|---|---|---|---|---|---|---|---|---|---|---|---|---|---|---|---|
| 11 | 126 | 14 | 17 | 14 | 11 | 36 | 2 | 1 | 19 | 4 | 39 | 12 | 70 | 5 | 91 | 62 | 83 | 25 | 32 | 27 |

| 19 | 20 | 21 | 22 | 23 | 24 | 25 | 26 | 27 | 28 | 29 | 30 | 31 | 32 | 33 | 34 | 35 | 36 | 37 | 38 | 39 |
|---|---|---|---|---|---|---|---|---|---|---|---|---|---|---|---|---|---|---|---|---|
| 18.5 | 12.3 | 16.4 | 15.7 | 16.9 | 20 | 20.3 | 10.5 | 16 | 12.1 | 20.2 | 17.4 | 14 | 15.6 | 21.8 | 18.3 | 17.6 | 15.8 | 14.6 | 18.2 | 17.4 |
| 10.3 | 8.4 | 10.1 | 7.6 | 8.7 | 10.5 | 12.9 | 8 | 12 | 9.4 | 9.5 | 8.5 | 7.1 | 8.2 | 12.4 | 11.2 | 11.4 | 8.2 | 7.4 | 7.4 | 8.8 |
| 13.3 | 7.6 | 9.4 | 11.6 | 11 | 14.5 | 12.2 | 7 | 11 | 7.9 | 12.8 | 10 | 10.8 | 10 | 14.8 | 11.9 | 10.5 | 8 | 8.4 | 11.5 | 9.7 |
| 42.4 | 28.2 | 35.9 | 34.9 | 36.6 | 45 | 45.5 | 25.5 | 39 | 29.4 | 42.5 | 35.9 | 32.5 | 33.8 | 49 | 41.3 | 39.6 | 32 | 30.5 | 37.5 | 36.6 |

| 19 | 20 | 21 | 22 | 23 | 24 | 25 | 26 | 27 | 28 | 29 | 30 | 31 | 32 | 33 | 34 | 35 | 36 | 37 | 38 | 39 |
|---|---|---|---|---|---|---|---|---|---|---|---|---|---|---|---|---|---|---|---|---|
| 4 | 6.3 | 5.6 | 7.4 | 5.3 | 4.9 | 3.8 | 6.5 | 10 | 5.1 | 6.2 | 4.9 | 7.2 | 5.1 | 4.2 | 5.4 | 6.3 | 4.8 | 3.7 | 7.4 | 5.3 |
| 1.5 | 4.4 | 1.5 | 2.4 | 1.4 | 0.8 | 1.9 | 1 | 4 | 1.4 | 3.5 | 1.8 | 5.6 | 2.8 | 3.2 | 1.9 | 3 | 4.7 | 2.7 | 2.4 | 3.9 |
| 5.6 | 6.4 | 3.9 | 6.5 | 5.6 | 5.4 | 3.2 | 8.5 | 0 | 3.1 | 3.5 | 4.6 | 6.4 | 3 | 5.2 | 4.5 | 6.1 | 3.3 | 3.5 | 8.6 | 4.3 |
| 6 | 8 | 6.1 | 9.8 | 9.1 | 6.5 | 4.4 | 8 | 6 | 6 | 5.5 | 5.8 | 7 | 3.6 | 8.8 | 5 | 7.2 | 4.7 | 6.4 | 9.1 | 6 |
| 9.4 | 12.3 | 7.9 | 12.5 | 9.9 | 8.2 | 7.1 | 11.5 | 13 | 5.7 | 7.2 | 7.7 | 11.9 | 5.1 | 11.8 | 6.8 | 11.4 | 7.7 | 7.5 | 13.9 | 8.9 |
| 3.5 | 4.8 | 2.5 | 3.4 | 3.6 | 4.7 | 3.1 | 5.5 | 6 | 2.9 | 5 | 4.7 | 5.7 | 4 | 4.4 | 3.9 | 4.6 | 3.4 | 3.4 | 7.2 | 5.6 |
| 30 | 42.2 | 27.6 | 41.9 | 34.9 | 30.5 | 23.7 | 41 | 39 | 24.1 | 31 | 29.6 | 43.8 | 23.6 | 37.6 | 27.5 | 38.7 | 28.6 | 27.2 | 48.6 | 33.8 |

| 19 | 20 | 21 | 22 | 23 | 24 | 25 | 26 | 27 | 28 | 29 | 30 | 31 | 32 | 33 | 34 | 35 | 36 | 37 | 38 | 39 |
|---|---|---|---|---|---|---|---|---|---|---|---|---|---|---|---|---|---|---|---|---|
| 24.3 | 27.1 | 26.1 | 22.6 | 21.8 | 26.5 | 26.6 | 25 | 23 | 16.5 | 32 | 24.2 | 23.3 | 23.3 | 26.4 | 19.7 | 18.7 | 25 | 15.8 | 24.8 | 20.8 |

| 19 | 20 | 21 | 22 | 23 | 24 | 25 | 26 | 27 | 28 | 29 | 30 | 31 | 32 | 33 | 34 | 35 | 36 | 37 | 38 | 39 |
|---|---|---|---|---|---|---|---|---|---|---|---|---|---|---|---|---|---|---|---|---|
| 9.1 | 22.2 | 7.1 | 17.6 | 21.4 | 9.1 | 27.8 | 0 | 0 | 15.8 | 50 | 15.4 | 0 | 5.7 | 0 | 7.7 | 6.5 | 10.8 | 12 | 21.9 | 33.3 |
| 0 | 3.2 | 0 | 0 | 0 | 0 | 2.8 | 0 | 0 | 5.3 | 0 | 5.1 | 8.3 | 4.3 | 0 | 1.1 | 0 | 2.4 | 0 | 3.1 | 3.7 |
| 9.1 | 7.1 | 7.1 | 0 | 0 | 0 | 5.6 | 50 | 0 | 0 | 0 | 2.6 | 0 | 4.3 | 0 | 1.1 | 1.6 | 4.8 | 4 | 9.4 | 11.1 |

| 19 | 20 | 21 | 22 | 23 | 24 | 25 | 26 | 27 | 28 | 29 | 30 | 31 | 32 | 33 | 34 | 35 | 36 | 37 | 38 | 39 |
|---|---|---|---|---|---|---|---|---|---|---|---|---|---|---|---|---|---|---|---|---|
| 0 | 15.9 | 0 | 5.9 | 7.1 | 18.2 | 0 | 0 | 0 | 26.3 | 0 | 2.6 | 45.5 | 21.4 | 0 | 17.6 | 17.7 | 7.2 | 16 | 21.9 | 7.4 |
| 27.3 | 11.1 | 0 | 11.8 | 7.1 | 9.1 | 5.6 | 50 | 0 | 5.3 | 0 | 20.5 | 36.4 | 15.7 | 0 | 12.1 | 14.5 | 3.6 | 8 | 18.8 | 14.8 |
| 72.7 | 73 | 100 | 82.4 | 85.7 | 72.7 | 94.4 | 50 | 100 | 68.4 | 100 | 76.9 | 18.2 | 62.9 | 100 | 70.3 | 67.7 | 89.2 | 76 | 59.4 | 77.8 |

| 19 | 20 | 21 | 22 | 23 | 24 | 25 | 26 | 27 | 28 | 29 | 30 | 31 | 32 | 33 | 34 | 35 | 36 | 37 | 38 | 39 |
|---|---|---|---|---|---|---|---|---|---|---|---|---|---|---|---|---|---|---|---|---|
| 54.5 | 69 | 50 | 47.1 | 35.7 | 45.5 | 83.3 | 0 | 100 | 57.9 | 50 | 66.7 | 33.3 | 22.9 | 100 | 25.3 | 30.6 | 81.9 | 72 | 68.8 | 81.5 |
| 9.1 | 23 | 35.7 | 47.1 | 42.9 | 18.2 | 30.6 | 0 | 0 | 5.3 | 25 | 15.4 | 8.3 | 20 | 60 | 11 | 19.4 | 30.1 | 12 | 40.6 | 18.5 |
| 54.5 | 31 | 42.9 | 64.7 | 57.1 | 54.5 | 50 | 50 | 0 | 47.4 | 50 | 23.1 | 58.3 | 55.7 | 100 | 37.4 | 30.6 | 36.1 | 48 | 40.6 | 33.3 |

**Table 2. Association table of the cluster-based genome-wide association study.**

| Cluster no. | Chr | hg19 | REF | ALT | ALT_FREQ | OR | 95% CI | *p*-value | Function | Gene symbol |
|---|---|---|---|---|---|---|---|---|---|---|
| 5 | 8 | 77348203 | C | T | 33.67% | 2.61 | (1.86–3.66) | $2.60 \times 10^{8}$ | ncRNA_intronic | LINC01111 |
| 7 | 7 | 13304197 | C | G | 1.90% | 15.46 | (5.94–40.23) | $2.00 \times 10^{8}$ | intergenic | ARL4A, ETV1 |
| 8 | 6 | 169690571 | A | G | 2.06% | 12.81 | (5.18–31.68) | $3.34 \times 10^{8}$ | intergenic | THBS2, WDR27 |
| 8 | 9 | 107163339 | G | A | 1.57% | 13.51 | (5.33–34.25) | $4.10 \times 10^{8}$ | intergenic | LOC105376194, OR13F1 |
| 9 | 12 | 125253297 | C | G | 1.10% | 59.17 | (13.86–252.6) | $3.59 \times 10^{8}$ | intergenic | NCOR2, SCARB1 |
| 14 | 14 | 83919384 | A | C | 1.76% | 13.53 | (5.31–34.44) | $4.72 \times 10^{8}$ | intergenic | LINC02301, SNORD3P3 |
| 16 | 9 | 3529465 | C | G | 2.67% | 8.53 | (4.05–17.96) | $1.71 \times 10^{8}$ | intergenic | RFX3, RFX3-AS1 |
| 17 | 7 | 7702162 | C | G | 2.30% | 7.43 | (3.64–15.18) | $3.68 \times 10^{8}$ | intronic | RPA3, UMAD1 |
| 19 | 5 | 44729840 | C | T | 1.84% | 38.83 | (10.91–138.2) | $1.61 \times 10^{8}$ | intergenic | LINC02224, MRPS30-DT |
| 19 | 5 | 45046992 | T | C | 1.70% | 49.39 | (12.73–191.7) | $1.74 \times 10^{8}$ | intergenic | MRPS30, HCN1 |
| 19 | 8 | 27064581 | A | G | 1.01% | 58.42 | (13.95–244.64) | $2.59 \times 10^{8}$ | intergenic | MIR548H4, STMN4 |
| 21 | 10 | 127919286 | G | A | 1.03% | 49.87 | (13.06–190.4) | $1.07 \times 10^{8}$ | intronic | ADAM12 |
| 23 | 15 | 67441517 | C | T | 1.11% | 84.58 | (22.43–318.94) | $5.64 \times 10^{11}$ | intronic | SMAD3 |
| 23 | 22 | 48720236 | G | T | 1.09% | 54.59 | (12.97–229.69) | $4.88 \times 10^{8}$ | intergenic | MIR3201, TAFA5 |
| 24 | 8 | 2803219 | C | A | 1.84% | 46.49 | (12.26–176.22) | $1.64 \times 10^{8}$ | intronic | CSMD1 |
| 24 | 9 | 116350142 | G | A | 1.34% | 46.83 | (12.35–177.52) | $1.54 \times 10^{8}$ | intronic | RGS3 |
| 24 | 17 | 15166354 | C | T | 1.75% | 44.82 | (11.57–173.67) | $3.74 \times 10^{8}$ | intronic | PMP22 |
| 25 | 1 | 42241824 | C | T | 2.01% | 10.99 | (4.66–25.92) | $4.40 \times 10^{8}$ | intronic | HIVEP3 |
| 25 | 14 | 32339580 | G | A | 1.49% | 13.82 | (5.49–34.79) | $2.52 \times 10^{8}$ | intergenic | NUBPL, LINC02313 |
| 28 | 1 | 186735538 | G | A | 1.04% | 30.26 | (9.01–101.57) | $3.42 \times 10^{8}$ | intergenic | PACERR, PLA2G4A |
| 28 | 3 | 155821228 | A | C | 1.15% | 40.14 | (11.62–138.63) | $5.26 \times 10^{9}$ | intergenic | GMPS, KCNAB1 |
| 31 | 7 | 147512932 | G | T | 1.96% | 32.64 | (9.64–110.51) | $2.13 \times 10^{8}$ | intronic | CNTNAP2 |
| 31 | 17 | 35386339 | T | C | 1.78% | 44.07 | (11.57–167.91) | $2.91 \times 10^{8}$ | intronic | AATF |
| 35 | 20 | 57142361 | G | T | 1.09% | 18.43 | (6.52–52.11) | $3.91 \times 10^{8}$ | ncRNA_intronic | APCDD1L-DT |
| 37 | 19 | 20047738 | T | G | 6.34% | 7.63 | (3.68–15.82) | $4.57 \times 10^{8}$ | intergenic | ZNF93, ZNF682 |
| 39 | 17 | 27641326 | G | A | 1.15% | 20.51 | (6.94–60.62) | $4.69 \times 10^{8}$ | intergenic | NUFIP2, MIR4523 |
| 39 | 17 | 63634041 | G | A | 1.14% | 25.40 | (8.02–80.46) | $3.84 \times 10^{8}$ | intronic | CEP112 |

REF, reference allele; ALT, alternative allele; ALT_FREQ, alternative allele frequency; OR: odds ratio; CI, confidence interval.

DPP10, DSCAM, FOXK1, GABBR2, GRIN2A5, ITPR1, NTM, SDK1, SNCA, SRRM4, and ZNF678. The proteins encoded by CNTNAP5 and ZNF678 exhibited CNTNAP and ZNF domains, respectively. In this study, proteins encoded by CNT-NAP2 in Cluster 30 and ZNF93 and ZNF682 in Cluster36 also contained the CNTNAP and ZNF domains, respectively.

## Discussion

It is implausible that decreasing the sample size would result in the identification of more significant signals in a GWAS. Therefore, we conducted the present follow-up study. The lack of reproducibility suggests that the cGWAS results may be false positives owing to some technical factors. However, the fact that several signals emerge following clustering and that many of the suggested gene regions are associated with ASD and related diseases may provide supporting evidence for cGWAS validity.

The first point is that the SSC study results were not entirely replicated. In the present and previous studies, only signals in genes encoding proteins with CNTNAP and ZNF domains were consistently observed. Although the variables used in the clustering were different, the limited replication strongly suggests that the signals emerging from the cGWAS approach may be false positives due to technical reasons rather than those inherent in the ASD subgroup.

This study has some limitations. First, although the SPARK cohort is one of the largest genetic cohorts focused on ASD, the limited statistical power of cGWAS to consider variants adequately due to the small sample size cannot be overlooked. It remains uncertain whether clusters with small sample sizes truly represent distinct groups. Moreover, interpreting odds ratios in clusters with insufficient cases was not feasible. Future clarification on the validity of these clusters may be possible with the availability of larger cohorts dealing with ASD and encompassing richer phenotypic information.

Second, the optimal selection of variables, algorithms, cluster numbers, and hyperparameters employed in this study remains unclear. Phenotypic variables chosen included DCDQ, SCQ, and RBS-R scores; age at initial registration in months; ethnicity; dominant hand; and history of medication, biomedical intervention, and intensive behavioral intervention. ASD manifests numerous other symptoms and characteristics [2,91,92]. Therefore, it is essential to carefully examine and narrow down a wide range of variables, considering their relationships in future studies. We selected the DEC algorithm [13,14] for its ability to simultaneously learn feature representations and cluster assignments using deep neural networks, representing one of the most contemporary techniques. Although DEC proves useful, the emergence of alternative algorithms is plausible in the future. A sensitivity analysis was conducted for the set number of clusters and hyperparameters. Although the optimality of the cluster number and hyperparameters in this study remains uncertain, our sensitivity analysis suggests a reasonable degree of validity in the methodology employed. Regarding the number of clusters, we observed that fewer clusters resulted in fewer detected variants, whereas increasing the number of clusters led to the detection of more variants. This is consistent with the hypothesis that cluster-specific variants can be identified by dividing patients with ASD into more homogeneous clusters based on various phenotypes. Changes in hyperparameters, such as batch size, did not significantly alter the results.

As indicated above, the results of the present study suggest that the signals emerging from the cGWAS approach may not be those originally possessed by ASD subgroups. However, some of our results indicate that, to a certain extent, cGWAS could potentially enhance the understanding of ASD pathogenesis. Compared with our previous study [11], two factors were replicated in the present study, which might indicate the validity of the cGWAS. First, several signals emerged due to clustering. This is somewhat consistent with a simulation study, suggesting that more signals are obtained by dividing into more homogeneous clusters [10], although it is unclear whether they were divided into more homogeneous clusters. Second, several gene regions have been suggested to be associated with ASD and related diseases. As suggested by previous studies, along with genes directly associated with ASD, we observed several other genes associated with ASD-related diseases or symptoms [33–36,37,39,40,42,46,48,49,51,57,59,60,62–66,70,71,73–75,77,79–81,83,88–90]. The ASD phenotype overlaps with other conditions, such as attention deficit hyperactivity disorder, epilepsy, mental retardation, Down's syndrome, schizophrenia, depressive symptoms, Parkinson's disease, Alzheimer's disease, Machado–Joseph's diseases, and post-traumatic stress disorders. Therefore, genes associated with these diseases were also identified in the current study. For example, some Parkinson's disease-related gene signals may be interpreted as follows: the presence of a certain gene mutation may be observed as an ASD-like symptom in childhood and diagnosed as ASD; however, with aging and cumulative exposure to environmental factors, symptoms may change slightly to a Parkinson's disease-like phenotype and be diagnosed as Parkinson's disease. Alternatively, ASD may not be diagnosed during childhood, but Parkinson's disease is diagnosed in old age. Sleep disturbances and microcephaly are frequently observed in patients with ASD. Dysregulation of G protein signaling, or mitochondrial dysfunction, has also been reported as an etiology of ASD [66,73]. Almost all statistically significant genes in the present study revealed using cGWAS were associated with ASD, its symptoms, and/or its pathways. These findings imply that clustering may be effective for identifying subgroups that share similar underlying disease causes.

Several statistically significant SNPs identified in the present study are reportedly associated with cancer. However, recent research has shown that there is a significant overlap between ASD and cancer risk genes [30–32,41,43,44,50,52–54,58,78,82,85–87]. For instance, ETV1 is associated with prostate cancer; RFX3-AS1, MRPS30, MRPS30-DT, and MIR3201 are associated with breast cancer; MIR548H4, AATF, and APCDD1L-DT are

**Table 3. Annotation of the genome-wide significant genes in the present cluster-based genome-wide association study primarily in relation to autism spectrum disorder.**

| Cluster no. | Chr | Gene symbol | Associated disease or affected status |
|---|---|---|---|
| 5 | 8 | LINC01111 | (Long Intergenic Non-Protein Coding RNA) |
| 7 | 7 | ARL4A | Glioma [30] |
| 7 | 7 | ETV1 | Prostate cancer [31], cranial germinomas [32] |
| 8 | 6 | THBS2 | Alzheimer's disease [33] |
| 8 | 6 | WDR27 | Sleep disturbance [34], Alzheimer's disease [35] |
| 8 | 9 | LOC105376194 | (Orthologs have not yet been determined) |
| 8 | 9 | OR13F1 | Olfactory receptors [36] |
| 9 | 12 | NCOR2 | Attention deficit hyperactivity disorder [37], spinal muscular atrophy [38], depression [39] |
| 9 | 12 | SCARB1 | Alzheimer's disease [40] |
| 14 | 14 | LINC02301 | (Long Intergenic Non-Protein Coding RNA) |
| 14 | 14 | SNORD3P3 | (non-annotated gene) |
| 16 | 9 | RFX3 | ASD [37], attention deficit hyperactivity disorder [37] |
| 16 | 9 | RFX3-AS1 | Breast cancer [41] |
| 17 | 7 | RPA3 | Machado–Joseph disease [42], glioma [43], gastric cancer [44] |
| 17 | 7 | UMAD1 | Seizure susceptibility [45] |
| 19 | 5 | HCN1 | Epilepsy [46], ASD [47], post-traumatic stress disorder [48], Parkinson's disease [49], Alzheimer's disease [49] |
| 19 | 5 | LINC02224 | (Long Intergenic Non-Protein Coding RNA) |
| 19 | 5 | MRPS30-DT | Breast cancer [50] |
| 19 | 5 | MRPS30 | Parkinson's disease [51], breast cancer [52] |
| 19 | 8 | MIR548H4 | Lung cancer [53], head and neck squamous cell carcinoma [54] |
| 19 | 8 | STMN4 | ASD [55,56] |
| 21 | 10 | ADAM12 | ASD [57], schizophrenia [57], brain cancer [57], Alzheimer's disease [57], Down syndrome [57] |
| 23 | 22 | MIR3201 | Breast and ovarian cancers [58] |
| 23 | 22 | TAFA5 | Depressive-like behaviors [59] |
| 23 | 15 | SMAD3 | Parkinson's disease [60] |
| 24 | 8 | CSMD1 | ASD [61], Alzheimer's disease [62], schizophrenia [63,64], Parkinson's disease [65] |
| 24 | 9 | RGS3 | Regulators of G protein signaling [66] |
| 24 | 17 | PMP22 | ASD [67], neuropathy [68] |
| 25 | 1 | HIVEP3 | ASD [69], Parkinson's disease [70,71], prostate cancer [72] |
| 25 | 14 | NUBPL | Mitochondrial disease [73], Parkinson's disease [74] |
| 25 | 14 | LINC02313 | (Long Intergenic Non-Protein Coding RNA) |
| 28 | 1 | PACERR | Epilepsy [75] |
| 28 | 1 | PLA2G4A | ASD [76], schizophrenia [77] |
| 28 | 3 | GMPS | Ovarian cancer [78], Alzheimer's disease [79] |
| 28 | 3 | KCNAB1 | Epilepsy [80,81], mental retardation [80] |
| 31 | 17 | AATF | Lung cancer [82], Alzheimer's disease [83] |
| 31 | 7 | CNTNAP2 | ASD [84] |
| 35 | 20 | APCDD1L-DT | Lung cancer [85] |
| 37 | 19 | ZNF93 | Ovarian cancer [86] |
| 37 | 19 | ZNF682 | Barrett's esophagus [87] |
| 39 | 17 | CEP112 | Attention deficit hyperactivity disorder [41] |
| 39 | 17 | NUFIP2 | Microcephaly [88] |
| 39 | 17 | MIR4523 | (non-coding micro-RNA) |

ASD, autism spectrum disorder.

associated with lung cancer. These types of cancer are strongly associated with ASD [93,94]. Regarding the genes involved in ASD and cancer, the list of cancer-associated genes identified in some clusters was almost identical to that of ASD-associated genes. Therefore, it is problematic to explain these findings solely in terms of false positives.

In addition to the two aforementioned replications, the fact that some agreement exists between the characteristics of the clusters and those inferred from gene expression may support the validity of cGWAS. The unique characteristics of each cluster obtained in this study have been illustrated in a heatmap (Table 1). For instance, Cluster 8 was associated with relatively lower DCDQ scores than those of other clusters but higher SCQ and RBS-R scores, with genes such as THBS2, WDR27, and ORF13F1 being associated with Alzheimer's disease. The characteristics of this cluster, wherein repetitive behavior and social communication deficits were more prevalent and motor skills were relatively preserved, are consistent with some features of Alzheimer's disease inferred from the functions of these genes. Meanwhile, Cluster 25 exhibited relatively lower RBS-R scores than those of the other clusters but higher DCDQ and SCQ scores, with HIVEP3 and NUBPL being associated with Parkinson's disease. The characteristics of this cluster were consistent with those of the disease features. In Cluster 7, no major characteristics were observed for DCDQ, SCQ, or RBS-R scores, whereas ARL4A and ETV were associated with cancer. The weakness of ASD characteristics in this cluster may be attributed to their association with cancer. Similar relationships were observed for other clusters. These results suggested an association between cluster characteristics and gene function; thus, the clusters obtained were functionally valid. However, it is crucial to note that the characteristics of clusters may not necessarily be recognizable by humans. Because artificial intelligence extracts features inherent in a combination of many variables, the clusters formed, although being more homogeneous, may not always be easily comprehensible to humans. In other words, artificial intelligence may uncover clusters that humans have not been able to discover. In the future, it will be necessary to define the clusters discovered by artificial intelligence.

Therefore, it is difficult to ascertain whether the present study identified a definitive subgroup, and it would be premature to draw conclusions regarding whether cGWAS has effectively elucidated the pathogenesis of ASD. However, this study revealed that similar to the SSC study, several signals emerge due to clustering, and many of the gene regions suggested herein are associated with ASD and related diseases. Therefore, completely ruling out cGWAS may be unwarranted, and further research is required. We plan to apply cGWAS to other datasets of ASD and other diseases. It may not be entirely futile for more researchers to conduct genetic searches based on cGWAS or ASD subgrouping. In doing so, ensuring reproducibility across cohorts, and using interpretable models less prone to overfitting is paramount, especially in small cohorts, for applying robust modeling strategies that account for heterogeneity. Moreover, for GWASs that already result in a large number of signals, it may be possible to determine whether the signals can be separated by dividing them into clusters. ASD subgroup identification using a proper classification may be important. This may partially lead to the development of precision medicine for ASD and other multifactorial diseases.

## Acknowledgments

The authors would like to thank the families at SPARK as well as the staff at SFARI. We are also grateful to Shoji Tanaka and Kei Takahashi for their assistance with this study.

## Author contributions

**Conceptualization:** Fumihiko Ueno, Ippei Takahashi, Hisashi Ohseto, Shinichi Kuriyama.

**Funding acquisition:** Fumihiko Ueno, Shinichi Kuriyama.

**Methodology:** Fumihiko Ueno, Ippei Takahashi, Hisashi Ohseto, Tomomi Onuma, Akira Narita, Taku Obara, Mami Ishikuro, Keiko Murakami, Aoi Noda, Fumiko Matsuzaki, Hirohito Metoki, Gen Tamiya, Shigeo Kure, Shinichi Kuriyama.

**Project administration:** Fumihiko Ueno, Taku Obara.

**Supervision:** Shinichi Kuriyama.

**Visualization:** Fumihiko Ueno.

**Writing – original draft:** Fumihiko Ueno, Ippei Takahashi, Hisashi Ohseto, Shinichi Kuriyama.

**Writing – review & editing:** Fumihiko Ueno, Ippei Takahashi, Hisashi Ohseto, Tomomi Onuma, Akira Narita, Taku Obara, Mami Ishikuro, Keiko Murakami, Aoi Noda, Fumiko Matsuzaki, Hirohito Metoki, Gen Tamiya, Shigeo Kure, Shinichi Kuriyama.

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
