## [Decision Letter · Decision Letter 0]

13 Jan 2025

PONE-D-24-31453Deep-embedded clustering by relevant scales and genome-wide association study in autismPLOS ONE

Dear Dr. Kuriyama,

Thank you for submitting your manuscript to PLOS ONE. After careful consideration, we feel that it has merit but does not fully meet PLOS ONE’s publication criteria as it currently stands. Therefore, we invite you to submit a revised version of the manuscript that addresses the points raised during the review process.

**Kindly address the reviewer comments below. Particularly, applying covariates to the model and checking for independent signals are paramount to make it a comprehensive scientific study.**

We look forward to receiving your revised manuscript.

Kind regards,

Mohith Manjunath, Ph.D.

Academic Editor

PLOS ONE

**Journal Requirements:**

The present study was supported by the Ministry of Education, Culture, Sports, Science and Technology (MEXT) KAKENHI Grant Numbers 19H03894, 21K17304 and 22H03346. The funders had no role in study design, data collection and analysis, decision to publish, or preparation of the manuscript.

3. In the online submission form, you indicated that All data used in this study are available only to those who are granted access by the Simons Foundation. The datasets analyzed during the current study available from the corresponding author on reasonable request.

Reviewers' comments:

Reviewer's Responses to Questions

**Comments to the Author**

1. Is the manuscript technically sound, and do the data support the conclusions?

Reviewer #1: Partly

2. Has the statistical analysis been performed appropriately and rigorously? 

Reviewer #1: No

3. Have the authors made all data underlying the findings in their manuscript fully available?

Reviewer #1: No

4. Is the manuscript presented in an intelligible fashion and written in standard English?

Reviewer #1: Yes

5. Review Comments to the Author

**Reviewer #1:**  This is an interesting paper; however, can the authors please address:

1) How did the authors define “ethnicity” in the cluster analysis? Was it based on genetic data or self-reported information?

2) It is unclear what covariates were used in the GWAS analysis. At a minimum, covariates such as sex, age, and genetic principal components (PCs) should be included. This omission may increase the risk of false positives in the cGWAS results.

3) Did the authors assess linkage disequilibrium (LD) between the 90 identified loci? I suspect that several loci may be in strong LD. If so, please report only the independent signals. A conditional GWAS analysis could also help verify this.

6. PLOS authors have the option to publish the peer review history of their article (what does this mean? ). If published, this will include your full peer review and any attached files.

**Do you want your identity to be public for this peer review?** For information about this choice, including consent withdrawal, please see our Privacy Policy .

Reviewer #1: No

---

## [Author Response · Author response to Decision Letter 1]

28 Feb 2025

Response to Reviewers document uploaded in Attach Files section.

---

## [Decision Letter · Decision Letter 1]

27 Mar 2025

Deep-embedded clustering by relevant scales and genome-wide association study in autism

PONE-D-24-31453R1

Dear Dr. Kuriyama,

We’re pleased to inform you that your manuscript has been judged scientifically suitable for publication and will be formally accepted for publication once it meets all outstanding technical requirements.

Kind regards,

Mohith Manjunath, Ph.D.

Academic Editor

PLOS ONE

Additional Editor Comments (optional):

Reviewers' comments:

Reviewer's Responses to Questions

**Comments to the Author**

1. If the authors have adequately addressed your comments raised in a previous round of review and you feel that this manuscript is now acceptable for publication, you may indicate that here to bypass the “Comments to the Author” section, enter your conflict of interest statement in the “Confidential to Editor” section, and submit your "Accept" recommendation.

Reviewer #1: All comments have been addressed

2. Is the manuscript technically sound, and do the data support the conclusions?

Reviewer #1: Yes

3. Has the statistical analysis been performed appropriately and rigorously? 

Reviewer #1: Yes

4. Have the authors made all data underlying the findings in their manuscript fully available?

Reviewer #1: Yes

5. Is the manuscript presented in an intelligible fashion and written in standard English?

Reviewer #1: (No Response)

6. Review Comments to the Author

Reviewer #1: The authors have satisfactorily addressed all my concerns. I recommend it to be published in PLOS ONE now.

7. PLOS authors have the option to publish the peer review history of their article (what does this mean? ). If published, this will include your full peer review and any attached files.

**Do you want your identity to be public for this peer review?** For information about this choice, including consent withdrawal, please see our Privacy Policy .

Reviewer #1: No

---

## [Editor Report · Acceptance letter]

PONE-D-24-31453R1

PLOS ONE

Dear Dr. Kuriyama,

I'm pleased to inform you that your manuscript has been deemed suitable for publication in PLOS ONE. Congratulations! Your manuscript is now being handed over to our production team.

Kind regards,

on behalf of

Dr. Mohith Manjunath

Academic Editor

PLOS ONE